# miR-199a: A Tumor Suppressor with Noncoding RNA Network and Therapeutic Candidate in Lung Cancer

**DOI:** 10.3390/ijms23158518

**Published:** 2022-07-31

**Authors:** Wei Meng, Yanli Li, Binshu Chai, Xiaomin Liu, Zhongliang Ma

**Affiliations:** Lab for Noncoding RNA & Cancer, School of Life Science, Shanghai University, Shanghai 200444, China; 1418103598@shu.edu.cn (W.M.); liyanli@shu.edu.cn (Y.L.); chaibinshu@shu.edu.cn (B.C.); liuxiaom@sh.tobacco.com.cn (X.L.)

**Keywords:** miR-199a, non-coding RNAs, lung cancer, interplay, drug resistance, glucose metabolism

## Abstract

Lung cancer is the leading cause of cancer death worldwide. miR-199a, which has two mature molecules: miR-199a-3p and miR-199a-5p, plays an important biological role in the genesis and development of tumors. We collected recent research results on lung cancer and miR-199a from Google Scholar and PubMed databases. The biological functions of miR-199a in lung cancer are reviewed in detail, and its potential roles in lung cancer diagnosis and treatment are discussed. With miR-199a as the core point and a divergence outward, the interplay between miR-199a and other ncRNAs is reviewed, and a regulatory network covering various cancers is depicted, which can help us to better understand the mechanism of cancer occurrence and provide a means for developing novel therapeutic strategies. In addition, the current methods of diagnosis and treatment of lung cancer are reviewed. Finally, a conclusion was drawn: miR-199a inhibits the development of lung cancer, especially by inhibiting the proliferation, infiltration, and migration of lung cancer cells, inhibiting tumor angiogenesis, increasing the apoptosis of lung cancer cells, and affecting the drug resistance of lung cancer cells. This review aims to provide new insights into lung cancer therapy and prevention.

## 1. Introduction

Lung cancer is one of the most common cancers and one of the top five causes of cancer death both in China [1] and the United States [2]. The American Cancer Society (ACS) cancer statistics for 2022 showed that lung cancer has the second-highest incidence and the highest death rate. In addition, the GLOBOCAN 2020 estimates of cancer incidence and mortality produced by the International Agency for Research on Cancer [3] showed that, globally, the incidence of lung cancer is second only to breast cancer, and the mortality rate of lung cancer is first in the world. Lung cancer [4,5] can be preliminarily divided into two types: small cell lung cancer (SCLC) and non-small cell lung cancer (NSCLC). Small cell lung cancer accounts for about 15% of cases, and the vast majority of lung cancers are NSCLC, accounting for about 85%. NSCLC can be further classified into three groups: adenocarcinoma, squamous cell carcinoma, and large cell carcinoma. Adenocarcinoma is the most common type, accounting for approximately 50% of NSCLC cases, squamous carcinoma for 25–30%, and large cell carcinoma for 10–15% [6].

Proteins represent the basic function and end product of genetic information, yet less than 2% of the genome encodes proteins. The remaining transcripts are called non-coding RNAs (ncRNAs) [7]. In general, there are four classes of ncRNAs: tRNA-derived fragments (tRFs), long non-coding RNAs (lncRNAs), circular RNAs (circRNAs), and microRNAs (miRNAs). With the advances in research, it was recognized that ncRNAs were no longer “junk” transcripts, but important functional regulators that mediate various cellular biological processes, including chromatin remodeling, transcription, post-transcriptional modification and signaling transduction [8]. The regulatory network involving ncRNAs can affect many downstream target molecules, thereby driving specific cellular biological responses. Thus, ncRNAs act as key regulators of physiological programs of disorders, including cancer, such as leukemia [9], and cardiovascular disease [10]. An in-depth understanding of the complex regulatory network of ncRNAs may provide new treatment ideas and therapeutic interventions for major cancers, including lung cancer.

## 2. Materials and Methods

A literature search was carried out in the PubMed, Web of Science, and Google Scholar databases (until 15 June 2022) using the following research terms: “NSCLC”, “SCLC”, “miR-199a-3p”, “miR-199a-5p”, “drug resistance”, “MDR”, “methylation”, “signaling pathway”, “biomarkers”, “tumor angiogenesis”, “glucose metabolism in lung cancer”, “miRNAs”, “miR-199a interplay”, “lncRNAs”, “circRNAs” “tRFs”, “diagnosis of lung cancer”, “treatment of lung cancer”, “Extracellular vesicles”, “EV-ncRNAs” “targeted therapy”, “Radiotherapy and chemotherapy” and “RNA therapy”.

According to the tentative review structure and title, the above keywords were searched in the literature database individually or in pairs, mainly searched in PubMed. The studies from the search was carefully studied to obtain the themes and research conclusions of the articles, and they were organized and classified. Considering the limited collection of literature in a single literature database, Web of Science and Google Scholar database were used again to search, and the gaps were filled for improvement. During the whole process, only the studies in the English language and relevant Chinese articles published by the author’s laboratory were selected, and studies in other languages, such as Spanish, German, French, etc., were excluded. Analyzed articles included reviews, clinical trials (CT), real-life studies (RLS), and reports. The figures in the review were drawn using Adobe Photoshop CC 2018 software, and the tables were drawn using Microsoft Office Word. This review is based on previously conducted studies and does not contain any studies with human participants or animals performed by any of the authors.

## 3. Result and Discussion

### 3.1. Biogenesis of miR-199a

miR-199 is a highly conserved miRNA family, with two members: miR-199a and miR-199b. Currently, there are two types of pre-miRNAs: pre-miR-199a-1 (MI000242) and pre-miR-199a-2 (MI0000281), derived from chromosomes 19 and 1, respectively [11]. Chromosome 1 is wrapped around more than 245.52 million nucleotide base pairs and contains about 8% of the DNA in human cells. Chromosome 19 contains approximately 63 million base pairs and accounts for 2 to 2.5 percent of all DNA in cells. miR-199a-1 is transcribed from Chromosome 19, NC_000019.10 (10817426..10817496, complement) and its nucleotide sequence is G C C A A C C C A G T G T T C A G A C T A C C T G T T C A G G A G G C T C T C A A T G T G T A C A G T A G T C T G C A C A T T G G T T A G G C. miR-199a-2 is transcribed from Chromosome 1, NC_000001.11 (172144535..172144644, complement) and its nucleotide sequence is: A G G A A G C T T C T G G A G A T C C T G C T C C G T C G C C C C A G T G T T C A G A C T A C C T G T T C A G G A C A A T G C C G T T G T A C A G T A G T C T G C A C A T T G G T T A G A C T G G G C A A G G G A G A G C A. After the cleavage of miR-199a by Dicer, two kinds of mature miRNAs could be obtained from the 5′ arm and 3′ arm, respectively: miR-199a-5p (MIMAT0000231) and miR-199a-3p (MIMAT0000232) (Figure 1).

Studies [12,13] have shown that two mature types of miR-199a regulate the activities of normal cells and participate in the corresponding physiological or pathological processes. For example, miR-199a-5p regulates the differentiation of the human smooth muscle cell phenotypes [12]. miR-199a-3p promotes the proliferation of cardiomyocytes [14], etc. In addition, miR-199a-3p and miR-199a-5p could jointly target *Brm* [15] and form a double-negative feedback regulatory loop through Egr1 (Early Growth Response 1), resulting in the generation of two different cell types in the process of human carcinogenesis. Furthermore, miR-199a-3p and miR-199a-5p could also simultaneously target *ITGA3* (integrin α3) [16] and inhibit the migration and invasion of head and neck cancer cells. This shows that miR-199a-5p and miR-199a-3p are not only highly similar in terms of biosynthesis but also have surrogate and synergistic properties in performing biological functions.

### 3.2. Biological Roles of miR-199a in Lung Cancer

As one of the most important miRNA family members, miR-199a has been reported to be involved in various tumors as a suppressor or promoter. For example, miR-199a is underexpressed in ovarian cancer [17] (OC) and esophageal squamous cell carcinoma (ESCC) [18] and acts as a tumor growth inhibitor. However, in some cancer types, such as gastric cancer [19] and breast cancer [20], miR-199a-3p is highly expressed and is generally considered an oncomiRNA. Lung is the organ with the highest incidence of malignant tumor metastasis. Tumors of all tissues and organs in the body can basically metastasize to lung. Brown M et al. [21]. found that tumor cells could infiltrate the lymph node parenchyma, invade blood vessels, and seed lung metastases. For example, ESCC has been associated with pulmonary metastasis. OC lung metastasis is the most common metastatic symptom, often occurring in the late stage of OC. Patients mainly show symptoms such as a cough and persistent lung infection, which may eventually develop into lung cancer.

As we have seen, many cancer types are involved in the aberrant expression of miR-199a, which leads us to focus on miR-199a alone. In addition, the lung being the most common site of distant metastasis for these above malignancies makes us wonder about the biological role miR-199a plays in the development and progression of lung cancer. Recently, there has been increasing evidence that the aberrant expression of miR-199a is closely related to lung cancer, affecting its proliferation, apoptosis, autophagy, glucose metabolism, etc.

#### 3.2.1. Methylation and the Low Expression of miR-199a

Previously, miR-199a-5p was reported as a tumor suppressor in NSCLC, and the expression of miR-199a-5p was significantly decreased in NSCLC tissues and cell lines [22,23]. Still, the down-regulation mechanism of mir-199a-5p expression and the role of its downstream targeting factors in NSCLC are not fully understood. Yang et al. [24] showed that the methylation level of the miR-199a promoter was significantly higher in NSCLC tissues than in normal para-carcinoma tissue, and the methylation of the promoter led to a low level of miR-199a expression. Similarly, in solid tumors, including non-small cell lung cancer, Mudduluru G et al. [25] found that both miR-199a/b promoter regions were methylated through methylation-specific PCR, resulting in their low expression. The luciferase reporter gene experiment verified the targeting relationship between mir-199a/b and the Axl receptor. The low expression of key regulatory miRNAs caused by methylation rather than other factors affects the downstream target genes of miRNAs, and this mechanism exists in many cancer types [26,27].

In fact, there are few miRNAs with methylation marks similar to miR-199a, which may alter their stability and target recognition. Japanese scholar Konno M and his colleagues found that the m6A methylation modification of miRNAs profoundly altered the structure of RISC, including the structure around the RNA recognition site, thus affecting the recognition efficiency of target RNAs [28]. miRNAs were traditionally detected based on the assumption that they recognize and regulate target RNAs regardless of whether they are methylated or not, and their roles may actually change depending on their methylation status. The methylation level of miR-199a is significantly elevated in lung cancer tissues and cells compared to normal tissues. Therefore, methylated miR-199a is likely to be a biomonitoring marker for lung cancer and may be more promising and clinically valuable compared to established lung cancer biomarkers. Furthermore, a recent study [29] confirmed that circulating miRNA can be used as a novel, predictable, and non-invasive biomarker. Hence, circulating [30] methylated miR-199a has great potential as a diagnostic marker for lung cancer. Given the available cancer biomarkers [31], the methylation of miRNAs may become an important component of future early cancer detection systems.

#### 3.2.2. miR-199a Regulates Cell Functions and Tumor Angiogenesis

Tumor cells have three distinctive essential characteristics: immortality, migratory nature, and loss of contact inhibition. In addition, tumor cells have many physiological, biochemical, and morphological characteristics that differ from normal cells. One of the critical biological functions played by miR-199a in lung cancer is regulating lung cancer cells functions. HIF-1α is a key regulator of hypoxia-induced cell proliferation. One study [32] found that miR-199a could inhibit hypoxia-induced NSCLC cell proliferation by targeting *HIF-1α*. Coincidentally, Yang et al. [33] confirmed the low expression of miR-199a-5p in NSCLC tissues and cell lines by qRT-PCR, and the overexpression of miR-199a-5p could inhibit the proliferation, migration, and invasion of NSCLC cells and increase cell apoptosis via targeting HIF-1α. Bai et al. [34] found that miR-199a-3p up-regulation could significantly inhibit the growth of NSCLC cells in vivo and promote mitochondria-mediated cell apoptosis. Subsequently, the targeting between miR-199a-3p and *ZEB1* (Zinc Finger E-box Binding Homeobox 1) was verified by a dual-luciferase reporting assay. miR-199a-3p suppressed the viability and proliferation of NSCLC cells via targeting CHML (choroideremia-like) cells, which were highly expressed in tissues and cell lines of NSCLC, and promoted the proliferation of NSCLC cells [35]. The dual-luciferase reporting assay confirmed that *CHML* was the target of miR-199a-3p. Similarly, A-kinase anchoring protein 1 (AKAP1) was a downstream target gene of miR-199a-5p, and miR-199a-5p overexpression inhibited proliferation and invasion of NSCLC cells [24], while AKAP1 overexpression partially rescued the malignant phenotype of NSCLC cells. In addition, studies found [36] that miR-199a-3p overexpression inhibited the growth of lung adenocarcinoma (LUAD) cells in vitro and in vivo and increased cell apoptosis.

Our lab [23] reported that miR-199a-5p plays a tumor suppressor role in NSCLC by directly targeting the *MAP3K11* gene. In addition, the increased expression level of miR-199a-5p could inhibit the proliferation of lung cancer cells and arrest the cell cycle in the G1 phase. In another of our research papers [37], we found that miR-199a-3p/5p were down-expressed in NSCLC tissue samples, cell lines, and the patient sample database. miR-199a-3p/5p overexpression could significantly suppress cell proliferation, migration ability and promote apoptosis. Additionally, ras homolog enriched in the brain (Rheb) was identified as a common target of miR-199a-3p and miR-199a-5p, which participated in regulating mTOR signaling pathway. In addition, our findings reveal that miR-199a-3p/5p is shown to enhance the sensitivity of gefitinib to EGFR-T790M in NSCLC. These results prove that miR-199a-3p/5p can act as cancer suppressor genes to inhibit the mTOR signaling pathway by targeting *Rheb*, which in turn inhibits the regulatory process of NSCLC. Collectively, our research showed that agomiR-199a-3p and agomiR-199a-5p inhibited tumor growth, and miR-199a-5p and miR-199a-3p have a GISTIC anti-tumor effect.

Tumor angiogenesis is a ceaseless process that cannot be self-regulated and is of great biological importance to tumor cell growth and migration. After tumor vascularization, tumor cell growth enters an exponential phase, both in terms of volume and number. These over-proliferating tumor cells migrate to other tissue sites through lymphatic or blood circulation and form secondary tumors. Similarly, miR-199a regulates tumor angiogenesis in lung cancer. HIF-1α (hypoxia-inducible factors 1-α) is an important transcription factor regulating tumorigenesis. It has been confirmed that miR-199a-5p could inhibit the generation of tumor blood vessels by targeting *HIF-1α* [38]. miR-199a-3p is also involved in the generation and metastasis of tumor blood vessels in solid tumors [39].

As mentioned above, we can learn that miR-199a not only directly inhibits the proliferation and migration of tumor cells but also inhibits the angiogenesis of tumors at the “source”. Therefore, we can expect a good druggability and clinical application value of miR-199a in anti-tumor angiogenesis and this is full of potential. In recent years, new advances have been made in exploring the regulatory role of ncRNAs in tumor angiogenesis. For example, RNA sponges, specific interfering molecules targeting ncRNAs that are proto-oncogenes, were synthesized for anti-tumor angiogenesis therapy [40]. In conclusion, ncRNAs such as miR-199a, which regulate tumor angiogenesis, could be a target for the development of new drugs for the treatment of cancer. In the near future, the relationship between ncRNAs and angiogenesis will be better understood, and their value will provide original and potential strategies for cancer management. The results of the above study are shown in Table 1.

#### 3.2.3. miR-199a Suppresses the Progression of Lung Cancer via Mediating Signaling Pathway

The cellular processes that are dysregulated in cancer cover a wide range of cellular signaling. miR-199a is also involved in regulating these signaling pathways in lung cancer. For example, miR-199a-5p inhibited the STAT (signal transducer and activator of transcription) signaling pathway by targeting *HIF-1α*, ultimately inhibiting the progression of NSCLC [33]. The HIF-1 α/STAT3 axis inhibited the expression of miR-199a-5p, forming a positive feedback loop to promote the continuous progression of NSCLC. Similarly, the expression of miR-199a significantly decreased, while the expression of HIF-1α and VEGF (vascular endothelial growth factor) increased, in NSCLC *rats*. Both HIF-1α/VEGF signaling pathway inhibitors and miR-199a mimics significantly reduced HIF-1α and VEGF protein expression and inhibited cell proliferation [41]. This suggested that miR-199a inhibited the proliferation of NSCLC cells by targeting the HIF-1α/VEGF signaling pathway. Qi et al. [42] confirmed that PVT1 (plasmacytoma variant translocation 1) absorbed miR-199a, and miR-199a inhibited *caveolin1* expression, thus forming the PVT1/miR-199a/caveolin1 signaling pathway in lung cancer cells. The silencing of the signaling pathway significantly reduced the malignant phenotype of NSCLC cells. In addition, miR-199a-5p was found to inhibit the MAPK signaling pathway, thereby suppressing the proliferation of lung cancer cells [23]. The UPR (Unfolded Protein Response) signaling pathway has a close relationship with NSCLC, and study [22] has shown that miR-199a-5p can target *GRP78* (*HSPA5*, Gene ID: 3309), and thus regulate the UPR signaling pathway and ultimately affect the progression of NSCLC. miR-199a is also involved in cellular autophagy via signaling pathways. For example, miR-199a-5p blocked autophagy by activating the PI3K/Akt/mTOR signaling pathway and inhibiting the expression of autophagy-related proteins [43].

Currently, there is an urgent need to develop new drugs in the face of lung cancer, which has a high incidence and mortality rate. Given the limitations and inadequacies of existing cancer treatment drugs, we are interested in new target drugs that are highly effective, have few side effects, and have a low propensity to develop drug resistance. Thus, focusing our attention on these dysregulated signaling pathways in lung cancer is certainly a promising direction. In addition, the pharmacological properties, targeting efficiency, and binding ability of drugs need to be considered in the drug development process. In addition to directly targeting signaling pathways, ncRNAs such as miR-199a are an excellent alternative. The results of the above studies are summarized in Table 2.

#### 3.2.4. miR-199a Improves the Accuracy of Lung Cancer Diagnosis

The expression of miRNAs in the serum of newly treated lung cancer patients (LC), benign lung disease patients (PD) and healthy control group (HC) was detected by PCR chip. The 10 miRNAs in LC, including miR-199a-3p, were significantly higher than PD and HC in the biomarker verification period [44] (*p* < 0.05). A bioinformatics analysis suggested that the predicted targets of these miRNAs might be involved in cancer formation and development. Currently, low-dose computed tomography (LDCT) has been recommended as a routine screening for high-risk lung cancer patients, but LDCT has a high false positive rate. He et al. used the miRNAs panel that includes miR-199a-3p to aid in a CT scan [45]. The goal was to evaluate the diagnostic accuracy of LDCT imaging combined with the miRNAs panel in lung nodules, and the results show a significant reduction in false positives.

#### 3.2.5. miR-199a Regulates Glucose Metabolism in Lung Cancer

In 1924, Otto Warburg observed a curious phenomenon: tumor cells rapidly consumed glucose and converted it into lactic acid [46]. Previous studies [47,48,49] reported that inhibition of glycolysis and pentose phosphate pathways (PPP) could affect NSCLC cells’ growth. Cancer cells need glucose to generate energy through glycolysis and the tricarboxylic acid cycle [50]. Glycolysis rapidly provided ATP and PPP-produced NADPH, the former used for ribonucleotide synthesis and NADPH used for rapid proliferation [51,52,53]. The up-regulation of glycolysis levels has been observed in many cancers, including primary and metastatic cancers, and aerobic glycolysis is the most commonly used and preferred mechanism of glucose metabolism in cancer cells [54]. Previous studies [55,56,57,58,59,60,61] showed that signaling pathways such as Akt, Myc, ERK, and NF-κB play an important regulatory role during glycolysis. In NSCLC, high levels of glucose transporter 1 (GLUT1) and hexokinase 2 (HK2) promoted glucose uptake by lung cancer cells, which was the initial step of glucose metabolism [62,63]. Phosphofructokinase (PFK) is a key enzyme regulating glycolysis, and its up-regulation is one of the main characteristics of malignant tumors. Research [64] found that platelet-like PFK (PFKP) is highly expressed in lung cancer tissues and cell lines.

Ding et al. [65] found significant abnormalities in glucose metabolism, especially in the glycolysis pathway, in lung cancer cells. Through further metabonomics and WGCNA (Weighted Gene Co-expression Network analysis) combined analysis, it was found that glucose metabolism disorder was closely related to lung cancer, which might become a potential target for lung cancer treatment.

In our previous study, Xu et al. [66] found that SLC2A1, a member of the GLUTs, was a direct target of miR-199a-5p. Down-regulation of SLC2A1 suppressed NSCLC cell proliferation, consistent with the role of miR-199a-5p. Besides, the knockdown of SLC2A1 could suppress glycolysis. This study proved that miR-199a-5p suppressed NSCLC via targeting *SLC2A1*, which significantly maintained glucose homeostasis. Therefore, some miRNAs target key enzymes or transporters during glucose metabolism, and thus affect the glucose metabolism level of tumor tissues, a potential mechanism that deserves further investigation. In addition, as mentioned above, some signaling pathways regulate glucose metabolism levels, so some genes in these signaling pathways, as well as key enzymes and transporter proteins, can be targets for therapy. This deserves further study.

#### 3.2.6. miR-199a and Drug Resistance

Drug resistance often occurs during the treatment of lung cancer. Additionally, almost all approved drugs on the market face the problem of drug resistance [23,67,68]. Primary drug resistance is a phenomenon that already exists in tumor diagnosis, and acquired drug resistance is closely related to the selective pressure of treatment. The pharmacological properties of drugs, including their potency, binding affinity, and structural stability, largely influence the mechanisms of drug resistance that emerge during disease progression [69]. In recent years, more and more pieces of evidence [70] have shown that miRNAs are closely related to the survival and chemotherapy sensitivity of tumor patients, and the effect of miRNAs is closely related to the regulation of the expression of ABC (ATP-binding cassette) transporters. ABC transporters are regulated in different ways by miRNAs. This regulation can be divided into direct regulation (miRNAs directly bind to 3′UTR of transporter mRNA), indirect regulation (miRNAs influence factors of other regulatory transporters), and transcriptional-level regulation (miRNAs regulate promoters of transporter-coding genes) [71]. Furthermore, studies [72,73] reported that signal molecules, such as miRNAs, lncRNAs, and circRNAs, contained in exosomes were closely related to tumor drug resistance, which could promote tumor angiogenesis, immune escape, metastasis, and then mediate the drug resistance of tumor cells.

The NSCLC exhibited resistance to chemotherapeutic agents such as cisplatin [74] and doxorubicin (Dox) [75] during chemotherapy. The multidrug resistance (MDR) of tumors [76] was one of the leading causes of clinical chemotherapy failure, but the mechanism of drug resistance of tumors has yet to be fully elucidated. Currently, p-glycoprotein (P-GP), MDR-associated protein 1 (MRP1), and breast cancer drug resistance protein (BCRP) are recognized to be closely related to MDR [77]. In addition, studies have shown that extracellular vesicles (EVs) are involved in the communication between MDR cells and drug-sensitive cells, promoting the spread of the MDR phenotype. Sousa D et al. constructed an NSCLC model and identified miRNAs associated with the MDR phenotype by high-throughput sequencing. It was found [78] that these miRNAs can be selectively packaged into EVs and that these vesicles can transfer drug resistance to recipient drug-sensitive cells through humoral circulation. This study helps us better understand the impact of EVs on MDR. Except for EVs, autophagy activation is also involved in MDR. In SCLC lines (H446 and H69PR), the overexpression of miR-199a-5p increased the formation of autophagic lysosomes, P62, and autophagy-associated proteins (LC3II/LC3I), while the knockdown of miR-199a-5p only decreased the expression of P62. In contrast, in multidrug-resistant cell lines (H446/EP), the overexpression of miR-199a-5p only decreased P62. In conclusion, the miR-199a-5p/p62 axis regulates autophagy as a potential mechanism of cisplatin resistance in SCLC [43]. Zeng et al. [79] showed that miR-199a-5p expression was up-regulated in paclitaxel-resistant lung cancer cell lines (A549 and H1299). The low expression of miR-199a-5p induced autophagy and made cells re-sensitive to chemotherapy drugs, while the overexpression of miR-199a-5p inhibited autophagy and desensitized cells to various chemotherapy drugs. In addition, miR-199a-5p directly targeted P62 mRNA and reduced the expression level of P62. The loss of P62 might inhibit autophagy and induce MDR. The regulation of miR-199a-5p in autophagy may provide a new therapeutic strategy for future multidrug-resistant lung cancer treatment and drug development.

Moreover, the study of [80] showed that miR-199a-5p expression was significantly down-regulated in doxorubicin-resistant cell lines, and miR-199a-5p significantly increased the sensitivity of lung cancer cells to doxorubicin. Further studies showed that miR-199a-5p acts as a sensitizer by targeting *ABCC1* and *HIF-1α*, which are highly expressed in doxorubicin-resistant cells. Similarly, miR-199a-5p inhibited the development of lung cancer by suppressing HIF-1α and STAT3 and increased the sensitivity of lung cancer cells to bevacizumab. Based on these results, Lou et al. [81] subsequently proposed using exosomes as effective vectors of miR-199a-5p to transport it around hepatocellular carcinoma, thereby promoting the chemotherapy effect of adriamycin on cancer cells. These results provide ideas for promoting the sensitivity of NSCLC to chemotherapy drugs (Table 3). 

### 3.3. Interplay between miR-199a and Other ncRNAs 

A wide variety of cancers involve miR-199a, and the interactions between miR-199a and other ncRNAs deserve our attention. When ncRNAs regulate biological activities, circRNAs, tRFs, and lncRNAs can interact with miRNAs. At the same time, miRNAs can also react with tRFs and lncRNAs. lncRNAs and tRFs reversely affect other mRNAs and gene sequences [82]. In this process, miR-199a forms a vast regulatory network with downstream target mRNA as well as ncRNAs.

#### 3.3.1. miR-199a and lncRNAs

miR-199a can affect the half-life of lncRNAs by promoting their degradation, and lncRNAs can act as miR-199a “sponges” and reduce the regulatory effect of miR-199a in target mRNAs. LncRNA TPPO-AS1 (TMPO antisense RNA 1) was significantly up-regulated in retinoblastoma (RB) and negatively correlated with miR-199a-5p expression. Tppo-AS1 had a binding site for miR-199a-5p, thus negatively regulating the expression of miR-199a-5p, and Tppo-AS1 played a carcinogenic role in the occurrence and development of RB [83]. LncRNA LUCAT1 (lung cancer-associated transcription 1) was overexpressed in ovarian cancer cells [84] and contained a highly conserved miR-199a-5p binding site in the 3′-un-translation region (3′-UTR). By targeting miR-199a-5p, lncRNA LUCAT1 played a role in driving the malignant development of OC. In hepatocellular carcinoma (HCC), the overexpression of lncRNA 01133 promoted HCC cell proliferation [85], and further studies have shown that lncRNA 01133 increased SNAI1 (snail family transcriptional repressor 1) expression and induced EMT (Epithelial-mesenchymal transition) of HCC cells by sponging miR-199a-5p. In addition, hypoxia induces a significantly high expression of lncRNA NEAT1 (nuclear paraspeckle assembly transcript1) in HCC cells, which maintained the growth of HCC cells, inhibited apoptosis, and blocked the cell cycle. It was found that lncRNA NEAT1 was a competitive endogenous RNA (ceRNA) of miR-199a-3p/UCK2 (uridine-cytidine kinase 2) in HCC cells [86]. LncRNA DLX6-AS1 (long-chain noncoding growth stasis specific protein 6 antisense RNA1) was up-regulated in nasopharyngeal carcinoma tissues and cells [87]. The overexpression of DLX6-AS1 promoted the malignant phenotype of nasopharyngeal carcinoma cells and the expression of HIF-1α via targeting miR-199a-5p. In Ewing’s sarcoma tissues and cells, the down-regulation of lncRNA TUG1 (taurine up-regulated gene 1) or overexpression of miR-199a-3p could inhibit cell proliferation, migration, and invasion. TUG1, as a ceRNA, regulated the expression of *MSI2* (musashi2) by sponging miR-199a-3p, thereby promoting the malignant phenotype of Ewing’s sarcoma cells [88].

LncRNAs could also affect the progression of neurological diseases via interactions with miR-199a. LncRNA TUG1 expression was up-regulated in temporal lobe epilepsy (TLE). The knockout of TUG1 could enhance cell activity and inhibit cell apoptosis, and miR-199a-3p was a target of TUG1 [89]. Glioma is an invasive tumor from the nervous system, and more than 70% of primary malignant brain tumors are caused by glioma. LINC01140 expression was enhanced in glioma tissues. Further study [90] found that LINC01140 negatively regulated the expression of miR-199a-3p in gliomas, thereby increasing the expression of *ZHX1* (zinc fingers and homeoboxes 1) and promoting the development of gliomas. LncRNA ANRIL has been reported to play an important role in ischemic injury [91]. ANRIL up-regulated the expression of caveolin1 by sponging miR-199a-5p, thus activating the MEK/ERK pathway and protecting cells from ischemia-induced damage. The expression of miR-199a-3p was down-regulated in Parkinson’s disease, while the expression of lncRNA XIST (X-inactive specific transcript) was up-regulated. The overexpression of miR-199a-3p could inhibit apoptosis and promote cell proliferation. Further studies [92] showed that lncRNA XIST sponged miR-199a-3p and further accelerated the progression of Parkinson’s disease.

The above findings suggest that the interactions between lncRNAs and miRNAs affect a variety of major human diseases. In lung cancer, interactions between miR-199a and lncRNAs exist. The study of [93] showed that lncRNA 01123 increased the expression of C-MYC mRNA by sponging miR-199a-5p, thus promoting NSCLC cell proliferation and aerobic glycolysis. Currently, predicting the interactions between lncRNAs and miRNAs has become a popular direction. Zhang et al. [94]. proposed a sequence-derived linear neighborhood propagation method (SLNPM) to predict lncRNA-miRNA interactions. These findings can help us better understand the function of miR-199a (Table 4). From a point to a line to an entire regulatory network, a more macroscopic level may provide a new perspective for thinking about the treatment of various cancers, such as lung cancer.

#### 3.3.2. miR-199a and circRNAs

Studies in recent years have shown that circRNAs are common as sponges for miRNAs. CircRNAs activate downstream genes of miRNAs by competitively binding miRNAs. Similarly, miR-199a has many sponge circRNAs that form regulatory networks affecting the development of various cancers and diseases. CircVMA21 (circular RNAs vacuolar ATPase assembly factor) was decreased, while miR-199a-5p was up-regulated, in LPS-induced THP1 cells. The overexpression of circVMA21 inhibited LPS-mediated THP-1 cell viability, apoptosis, and inflammation by sponging miR-199a-5p [95]. The expression of circPVT1 was significantly reduced in TSPC (tendon stem/progenitor cell) cultured for a long time in vivo, and its up-regulation suppressed the aging process of TSPC and inhibited the self-renewal and migration of TSPC by sponging miR-199a-5p, thus weakening the negative regulation of miR-199a-5p on *SIRT1* expression [96]. In patients with colorectal cancer (CRC), circUBAP2 expression was up-regulated. Further studies [97] showed that circUBAP2 promoted the progression of CRC by sponging miR-199a and up-regulating VEGFA (vascular endothelial growth factor A). Compared with healthy ovarian tissue, circMUC16 was up-regulated in epithelial ovarian cancer (EOC) tissue and promoted autophagy, invasion, and metastasis of EOC cells by sponging miR-199a-5p [98]. Yang et al. [99] found that circ0060450 could act as a sponge for miR-199a-5p to release its target genes and further inhibit macrophage-mediated inflammation. Song et al. [100] found that circMTO1 could sponge miR-199a-3p to release its downstream target gene *PAWR* (PRKC Apoptosis WT1 Regulator Protein), thus inhibiting the occurrence and development of gastric cancer. Similarly, another circITCH found in GC could sponge miR-199a-5p and increase the expression of *Klotho* (a downstream target gene of miR-199a-5p), thereby inhibiting the metastasis of gastric cancer [101]. The expression level of circNRIP1 in OS tissues was significantly up-regulated, and the down-regulation of circNRIP1 could inhibit the proliferation and migration of OS (osteosarcoma) cells and promote cell apoptosis. A further bioassay showed [102] that circNRIP1 played a tumorigenic role in OS by sponging miR-199a. Further studies of such interactions (Table 5) may bring us surprises. After all, given the current research on circRNAs, including the mechanisms of formation and interactions and its mechanisms, further study or the development of corresponding cancer target drugs is not yet on the agenda.

#### 3.3.3. miR-199a and tRFs

In addition to the above lncRNAs and circRNAs, there is also the relationship between tRFs and miRNAs. TRFs are not random by-products of tRNA degradation but have precise sequence structures, specific expression patterns, and specific biological effects [103]. miRNAs sequences are found within tRFs [104]. For example, miR-1260a and miR-4521 sequences were found within tRF-3001a and tRF-1003, respectively [105]. Venkatesh T et al. [104] used miRbase to classify miRNAs that overlapped with tRFs sequences in humans. Twenty tRNA-derived miRNAs that shared sequences with tRFs were found. Among the 20 miRNAs, 5 miRNAs (miR-3182, miR-4521, miR-1260a, miR-1260b, and miR-7977) had significant predictive scores. In addition, the two types of ncRNAs perform similar functions in some diseases. For example, in acute pancreatitis (AP), Yang et al. [106] used bioinformatics methods to establish tRF-mRNA and miRNA-mRNA regulatory networks. The network revealed 29 central miRNAs and 19 central tRFs. GO analysis showed that the functions of the two networks were similar, mainly enriched in RNA splicing and mRNA processing. Although there are few direct interactions between miRNAs and tRFs at present, the above information indicates that there is indeed a relationship between the two.

Although the primary biological function of miRNAs is to perform post-transcriptional regulation of genes, as shown above, the regulation and interactions of miR-199a on other ncRNAs have reshaped our understanding of RNA biology. We can no longer simply assume that a single miRNA can regulate multiple targets. Instead, this must be expanded to incorporate the idea that miRNAs may regulate each other [107]. Moreover, miR-199a acts as a potent regulator and has been shown to drive oncogenic pathways or cancer-inhibiting pathways (with different background cancer types) [108], so the impact of interactions of potent miRNAs, including miR-199a, with other non-coding RNAs is profoundly valuable to study. Given the existing research tools and analytical practices, we are currently only at the “theoretical conclusion” stage. Further exploration of the mechanisms of interactions is necessary to help us better understand cancer, and thus develop new cancer therapies. Going forward, the impact of the miR-199a regulatory network must be noted in studies related to the role of miR-199 in lung cancer and beyond, as well as in its application in therapeutics. The role of miR-199a in lung cancer and the interplay between miR-199a and other ncRNAs are shown in Figure 2.

### 3.4. Discussion and Perspectives

#### 3.4.1. Recent Progress in Diagnosis and Treatment of Lung Cancer

The diagnosis of lung cancer is often delayed. This is mainly because the symptoms of lung cancer may be specific (confined to the lungs) or non-specific, and often the latter (non-specific). Therefore, most lung cancer patients are diagnosed as non-localized disease and the majority of NSCLC patients present at an advanced stage of disease [109]. Positive emission scanning (PET) and transbronchial (EBUS) or transesophageal ultrasound (EUS) biopsy techniques have improved mediastinal staging of the disease, and minimally invasive thoracoscopic surgery (VATS) has reduced postoperative complications and shortened hospital stays [110,111]. Recent low-dose computed tomography screening trials have shown good results in reducing mortality [112].

Surgical resection is the main and preferred method of treatment for lung cancer, but it is only applicable to patients with early-stage lung cancer [109]. Radiotherapy and chemotherapy are also the main means of fighting cancer, but their toxic side effects are too great [113,114]. In recent years, with the development of molecular biology, targeted therapy has become a new means of fighting cancer. Its molecular targeting, high specificity, targeted drugs, without blood and neurotoxicity and other characteristics, greatly improve the quality of life of patients [115]. Several clinical drugs have been developed [116,117,118], such as the inhibition of epidermal factor receptor (EGFR) [119], anaplastic lymphoma kinase (ALK) and ROS1, etc. Currently, several new treatment combinations and new drugs are being investigated. In conclusion, it is especially important to explore more key molecular markers for lung cancer, and finding possible therapeutic targets is the key to treat lung cancer.

#### 3.4.2. Prospects of ncRNAs for Diagnosis and Treatment of Lung Cancer

NcRNAs have high stability in body fluids, tissue-specific expression, and unique expression profiles, which make them excellent candidate biomarkers [120,121,122]. Patricia Le et al. pointed out that the detection of ncRNAs in EVs released by tumor cells by a less invasive body fluid biopsy is an excellent option for screening and therapeutic detection, and showed that studying the degree of dysregulation of ncRNAs can help develop ncRNA-based treatment [123], for example, by introducing exogenous inhibitors, siRNAs or overexpression vectors to restore or suppress abnormally expressed ncRNAs.

In addition, the subclass structure of extracellular vesicles (EVs) between cells and molecules, which mediates the transport of substances, including ncRNAs deserves further study. In recent years, studies have found elevated levels of specific subclasses of a broad range of noncoding RNAs (ncRNAs) in EVs, suggesting a highly selective and molecular regulatory process during the packaging of ncRNAs into EVs. It was verified that [124] miRNAs possess sorting sequences that determine their secretion in EVs or cellular retention. The ncRNA profiles of EVs were significantly different from those of donor cells, and some ncRNAs were significantly enriched in EVs. The identification of the expression profiles of ncRNAs in these extracellular vesicles can help us to better diagnose and prevent diseases including lung cancer and lung-related diseases. For example, some miRNAs (miR-21, miR-210, etc.) are significantly enriched in EVs in blood during lung infection and injury [125], and these miRNAs play important regulatory roles. Coincidentally, O’Farrell, H.E et al. demonstrated [126] that miRNAs in EVs from plasma have key biological information specific to lung cancer. Fifteen miRNAs, including miR-199a-5p, were differentially expressed between lung cancer patients and healthy non-smoking participants. Similarly, in liver fibrosis, Messner, C.J et al. identified four miRNA biomarkers including miR-199a-5p and highlighted their applicability as in vitro models for assessing miRNA release during fibrosis [127]. We hope to witness the same discoveries and breakthroughs in pulmonary fibrosis or other related diseases. These studies suggest that EV-ncRNAs may be developed as valuable biomarkers and therapeutic targets. Finally, due to the heterogeneity of tumors, a biological classification system consisting of a panel of biomarkers may be necessary to achieve a high sensitivity and specificity.

## 4. Conclusions

miR-199a is certainly a promising target and biomarker. Higher methylation levels of the promoter resulted in the low expression of miR-199a in lung cancer tissues and cell lines, but this is a detectable feature. Detecting its methylation level rather than the overall expression level can be better used for lung cancer diagnosis. After all, its expression is high in some lung cancer drug-resistant cell lines. Additionally, and this is a puzzling point, miR-199a expression is up-regulated in paclitaxel-resistant lung cancer cell lines [79] and down-regulated in doxorubicin-resistant lung cancer cell lines [80]. This may be because the two different drug treatments resulted in different biological responses and the activation of signaling pathways in the lung cancer cell lines, leading to different downstream targets of miR-199a regulation. This still needs further study, and like mathematics and physics, we need a standard unifying formula for more than just lung cancer and more than just miR-199a.

In addition, miR-199a participates in the regulation of multiple signaling pathways, acts as a regulatory molecule of tumor cell glucose metabolism, inhibits tumor angiogenesis, and directly affects a series of functions of lung cancer cells, which allows us to expect more from miR-199a. Indeed, growing evidence has shown [128] that miRNA-based therapies, either restoring or inhibiting miRNA expression and activity, hold great promise. Furthermore, miRNA mimics and miRNA inhibitors currently in preclinical development have shown promise as novel therapeutic agents [129]. Additionally, in these processes, HIF-α is an important downstream target and an important factor affecting the miR-199a regulation of lung cancer. HIF-α deserves our attention and may be an important breakthrough in elucidating the regulatory mechanism played by mir-199a in cancer, but further studies are still needed. miR-199a regulates cellular autophagy, which can affect drug resistance in lung cancer cells. In turn, autophagy may be a homeostatic process necessary to clear damaged organelles and restore macromolecules for cancer prevention [130]. So, in addition to focusing on drug resistance, this regulatory role should bring us new concerns, such as linking to cellular functions, focusing on autophagy-related signaling pathways, and understanding miRNA-regulated autophagic networks, which will be the basis for improving lung cancer treatment strategies. In addition, does miR-199a interact with proteins in addition to other ncRNAs? Additionally, what would be the effect of miR-199a’s direct interaction with the protein on cancer development and progression? After all, direct interactions between miRNAs and proteins have been shown to exist [131]. We look forward to seeing the full picture of the huge regulatory network of miR-199a with other ncRNAs, downstream target mRNAs, and proteins soon. This will undoubtedly be of great benefit to our research on the treatment of a range of diseases, including lung cancer and another way of thinking. Some of the identified target genes of miR-199a may also be the targets of future drugs. For example, as mentioned above, our lab studies found that miR-199a-3p and miR-199a-5p could directly co-target Rheb (Ras homolog enriched in the brain) in lung cancer [37], regulating the mTOR signaling pathway, inhibiting cell proliferation, migration, and promoting cell apoptosis. The same is true for our laboratory’s discovery of MAP3K11 as a target gene for miR-199a-5p in NSCLC [23]. This miRNA/target gene regulatory axis consisting of miR-199a and target genes is more thoroughly studied today, and we look forward to building on this foundation further toward R&D and clinical applications.

Currently, only a few ncRNAs have been studied in depth, such as miR-199a. Most of the functional biological mechanisms and biogenesis of ncRNAs are still unknown. Studies of ncRNAs lack homogeneity and standardization. Therefore, there is still a long way to go from discovery and research maturation to clinical application. In addition, clinical sequencing of patients with advanced lung cancer can be used to determine which oncogenes are mutated and is a more accurate diagnostic and prognostic tool that can better guide management and predict outcomes [132]. It can be used not only for effective targeted therapy but also to collect more clinical data for future studies. Differences and changes in transcript levels between normal cell tissues and corresponding cancer tissues [133] should also be taken seriously. Although the study of the tumor transcriptome is still in its early stages, future research could focus on those important substances that play a role in the development of cancer. Finally, whether as biomarkers for prevention, diagnosis, prognosis, or as therapeutic targets, ncRNAs have great applications as research hotspots for tumor therapy.

## Figures and Tables

**Figure 1 ijms-23-08518-f001:**
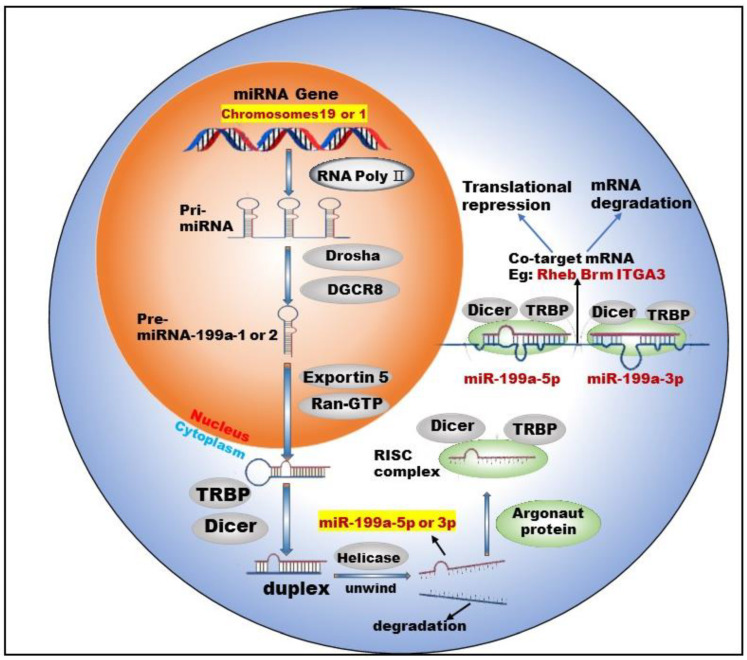
Biogenesis and co-targeting of miR-199a. The gene encoding miR-199a in the nucleus is processed into pri-miRNA catalyzed by RNA poly Ⅱ. The pri-miRNA is then processed into a pre-miRNA by the shearing action of a complex composed of Drosha (a ribonuclease enzyme) and DGCR8 (Human Recombinant Protein (P01)). Pre-miRNA exits the nucleus into the cytoplasm with the help of Exportin 5 (Exporting protein 5) and Ran-GTP (Ras-like nuclear protein, GTPase). In the cytoplasm, Dicer (a ribonuclease enzyme) and TRBP (TAR-RNA binding protein) further process the pre-miRNA by excising the stem–loop structure at the end to form an unstable double-stranded RNA (dsRNA). DsRNA is unwound into single-stranded DNA by the action of Helicase, where one single strand is degraded and the other becomes a mature miRNA. The mature miRNA forms RISC (RNA-induced silencing complex) with TRBP, Argonaut protein and Dicer enzymes to perform biological functions.

**Figure 2 ijms-23-08518-f002:**
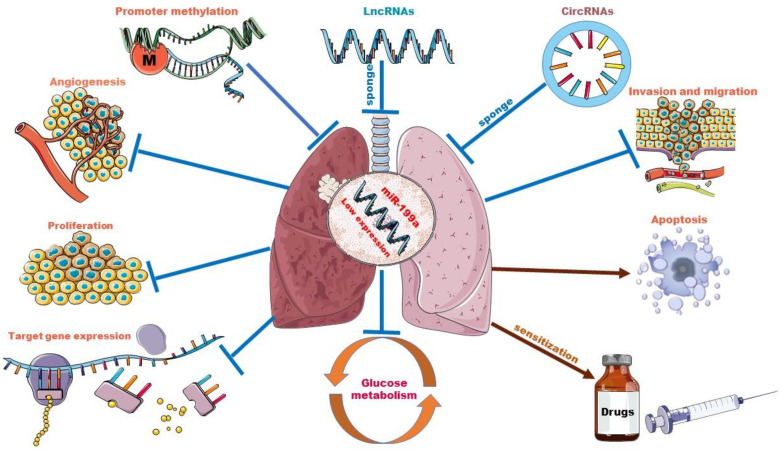
The role of miR-199a in lung cancer and interplay with other ncRNAs. Promoter methylation and sponge adsorption of lncRNAs and circRNAs resulted in low expression levels of miR-199a. miR-199a acts as a tumor suppressor in lung cancer by targeting downstream genes. Overexpression of miR-199a could inhibit lung cancer cell proliferation, infiltration and migration, glycolytic pathway and tumor angiogenesis and increase lung cancer cell apoptosis. In some drug-resistant lung tumor cell lines, it also plays a sensitizing role.

**Table 1 ijms-23-08518-t001:** miR-199a regulates cell functions and tumor angiogenesis in lung cancer.

	Biological Functions	Target	Ref
miR-199a-5p	Playing a tumor suppressor role in NSCLC	MAP3K11	[23]
miR-199a-5p	Inhibiting the proliferation, migration, and invasion of NSCLC cells and increasing cell apoptosis	HIF-1α	[33]
miR-199a-5p	Inhibiting proliferation and invasion of NSCLC cells	AKAP1	[24]
miR-199a-5p	Inhibiting the generation of tumor blood vessels	HIF-1α	[38]
miR-199a-3p	Inhibiting the growth of NSCLC cells in vivo and promoting mitochondria-mediated cell apoptosis	ZEB1	[34]
miR-199a-3p	Suppressing the viability and proliferation of NSCLC cells	CHML	[35]
miR-199a-3p	Inhibiting the growth of LUAD cells in vitro and in vivo and increasing cell apoptosis	AGR2	[36]
miR-199a-5p/3p	Suppressing cell proliferation, migration ability and promote apoptosis	Rheb	[37]

MAP3K11 (MAPK relevant potential target genes); Rheb (ras homolog enriched in brain); HIF-1α ((hypoxia-inducible factors 1-α); ZEB1 (Zinc Finger E-box Binding Homeobox 1); CHML (choroideremia-like); AKAP1 (A-kinase anchoring protein 1); ARG2 (anterior gradient 2).

**Table 2 ijms-23-08518-t002:** miR-199a is involved in the signaling pathway in lung cancer.

	Signaling Pathways Involved	Biological Functions	Ref
miR-199a-5p	STAT signaling pathway	Inhibiting the progression of NSCLC	[33]
miR-199a-5p	MAPK signaling pathway	Suppressing the proliferation of lung cancer cells	[23]
miR-199a-5p	UPR signaling pathway	Having an effect on ER stress, as well as a causative role in lung tumorigenesis	[22]
miR-199a-5p	PI3K/Akt/mTOR signaling pathway	Inhibiting the expression of autophagy-related proteins	[43]
miR-199a-5p/3p	mTOR signaling pathway	Acting as cancer suppressor genes	[37]

STAT (signal transducer and activator of transcription); VEGF (vascular endothelial growth factor); PVT1 (plasmacytoma variant translocation 1); MAPK (mitogen-activated protein kinase); UPR (Unfolded Protein Response).

**Table 3 ijms-23-08518-t003:** miR-199a regulates drug resistance.

	Biological Functions	Diseases/Cell Lines	Ref
miR-199a-5p	Increasing the formation of P62	In SCLC lines (H446 and H69PR)	[43]
miR-199a-5p	Decreasing the expression of P62	In multidrug-resistant SLCL cell lines (H446/EP)	[43]
miR-199a-5p	Inhibited autophagy and desensitized cells to a variety of chemotherapy drugs	In paclitaxel-resistant lung cancer cell lines (A549 and H1299)	[79]
miR-199a-5p	Increasing the sensitivity of lung cancer cells to doxorubicin	In doxorubicin-resistant lung cancer cell lines	[80]
miR-199a-5p	Increasing the sensitivity of lung cancer cells to bevacizumab	In NSCLC	[33]
miR-199a-5p/3p	Enhancing the sensitivity of gefitinib to EGFR-T790M	In NSCLC	[37]

**Table 4 ijms-23-08518-t004:** Interplay between miR-199a and lncRNAs.

	LncRNAs	Mechanism	Biological Effects	Diseases	Ref
miR-199a-5p	LncRNA TPPO-AS1 (TMPO antisense RNA 1)	LncRNA Tppo-AS1 has a binding site for miR-199a-5p	Negatively regulating the expression of miR-199a-5p	Retinoblastoma	[83]
miR-199a-5p	LncRNA LUCAT1 (lung cancer-associated transcription 1)	LncRNA LUCAT1 contains a highly conserved miR-199a-5p binding site in 3′-UTR	Driving the malignant development of OC	Ovarian cancer	[84]
miR-199a-5p	LncRNA 01133	LncRNA 01133 sponges miR-199a-5p	Inducing EMT	Hepatocellular carcinoma	[85]
miR-199a-5p	LncRNA DLX6-AS1 (long-chain noncoding growth stasis specific protein 6 antisense RNA1)	LncRNA DLX6-AS1 could target miR-199a-5p	DLX6-AS1 promoted the malignant phenotype of cells a via targeting miR-199a-5p	Nasopharyngeal carcinoma	[87]
miR-199a-5p	LncRNA ANRIL	LncRNA ANRIL sponges miR-199a-5p	Activating the MEK/ERK pathway	Ischemic injury	[91]
miR-199a-5p	LncRNA 01123	LncRNA 01123 sponges miR-199a-5p	Promoting NSCLC cell proliferation and aerobic glycolysis	NSCLC	[93]
miR-199a-3p	LncRNA NEAT1 (nuclear paraspeckle assembly transcript1)	LncRNA NEAT1 is a ceRNA of miR-199a-3p	Inhibiting the growth of HCC cells and increasing cell apoptosis	Hepatocellular carcinoma	[86]
miR-199a-3p	LncRNA TUG1 (taurine up-regulated gene 1)	LncRNA TUG1 sponges miR-199a-3p	Promoting the malignant phenotype of cells	Ewing’s sarcoma	[88]
miR-199a-3p	LncRNA TUG1	miR-199a-3p was a target of LncRNA TUG1	Enhancing cell activity and inhibiting cell apoptosis	Temporal lobe epilepsy	[89]
miR-199a-3p	LINC01140	LINC01140 could target miR-199a-3p	Increasing the expression of ZHX1	Gliomas	[90]
miR-199a-3p	LncRNA XIST (X-inactive specific transcript)	lncRNA XIST sponges miR-199a-3p	Accelerating the progression of Parkinson’s disease	Parkinson	[92]

**Table 5 ijms-23-08518-t005:** Interplay between miR-199a and circRNAs.

	CircRNAs	Mechanism	Biological Effects	Diseases/Cell Lines	Ref
miR-199a-5p	CircVMA21 (circular RNAs vacuolar ATPase assembly factor)	CircVMA21 sponges miR-199a-5p	Inhibiting cell viability, apoptosis, and inflammation	LPS-mediated THP-1	[95]
miR-199a-5p	CircPVT1	CircPVT1 sponges miR-199a-5p	Suppressing the aging process of TSPC	TSPC (tendon stem/progenitor cell)	[96]
miR-199a-5p	CircMUC16	CircMUC16 sponges miR-199a-5p	Promoting autophagy, invasion, and metastasis of cells	EOC (epithelial ovarian cancer)	[98]
miR-199a-5p	Circ0060450	Circ0060450 could act as a sponge for miR-199a-5p	Inhibiting macrophage-mediated inflammation	Type 1 Diabetes Mellitus	[99]
miR-199a-5p	CircITCH	CircITCH could sponge miR-199a-5p	Inhibiting the metastasis of gastric cancer	Gastric cancer	[101]
miR-199a-3p	CircMTO1	CircMTO1 could sponge miR-199a-3p	inhibiting the occurrence and development of gastric cancer	Gastric cancer	[100]

## Data Availability

All data included in this manuscript are from published scientific manuscripts and available on Google Scholar, Web of Science and PubMed (NIH).

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
