# Peer review of "miR-199a: A Tumor Suppressor with Noncoding RNA Network and Therapeutic Candidate in Lung Cancer"

_ijms, 2022, doi:10.3390/ijms23158518_

Round 1

Reviewer 1 Report

Dear Authors,

I carefully reviewed your article that aims to describe the miR-199a role in lung cancer and its possible therapeutic indications. After reviewing the manuscript I noticed that there are still major issues regarding its design and information presented in the body of the manuscript. The information is inaccurate, not structured and is leading to confusion. The manuscript needs professional English editing before a resubmission and reorganization. In this format is hard to follow and is of low interest. Please find below some of the minor and major points that should be addressed:

1.     Check the information in text for medical accuracy throughout the manuscript. For example the affirmation in line 485-486 regarding surgery in lung cancer is not true.

2.     Biogenesis of miR-199 needs to be better explained. What is the difference between chromosome 19 and 1? From where is this miRNA transcribed? More details are needed.

3.     Throughout the manuscript when you are discussing miR-199 functions you are mixing information between lung cancer and other cancers in a disproportionate manner which makes it difficult to follow. Information should be reorganized to highlight the roles in lung cancer.

4.     Chapter regarding miR-199a and signaling pathways is general and inaccurate. Try using a different title making it more specific.

5.     Line 10: ”The biological functions of miR-199a in lung cancer are reviewed in detail, and its possibility and prospects for diagnosing and treating of lung cancer are discussed.” – the underscored formulation is not the best; I recommend replacing it with its potential roles in lung cancer diagnosis and treatment are discussed.

6.     Line 13: ”which can better help us understand the mechanism of cancer occurrence and provide original therapeutic strategies. – the underscored formulations are not the best; I recommend replacing them with ”help us to better understand the” and ”provide the means for developing novel therapeutic strategies”.

7.     Line 14: ”In addition, the current methods of diagnosis and treatment of lung cancer are also reviewed.” – here, ”in addition” and ”also” are not both necessary.

8.     Line 16: ”including inhibiting the proliferation, infiltration, and migration of lung cancer cells, inhibiting tumor angiogenesis, increasing the apoptosis of lung cancer cells, and affecting the drug resistance of lung cancer cells.” – I recommend replacing ”including” with ”especially by”.

9.     Line 24: replace ”of death from cancer” with ”cancer death”.

10.  Line 36: since it is the first and only time you define ncRNAs, I think this explanation “Non-coding RNAs (ncRNAs), which do not encode proteins but perform biological 36 functions at the RNA level” should be more comprehensive.

11.  Line 38: as leukemia is actually a type of cancer, I don’t think it should be enumerated separately from cancer.

12.  Line 40: “In the process of life development, ncRNAs play an important regulatory role, especially in the occurrence and development of cancer.” – this formulation is slightly confusing. Firstly, “life development” is a very uncommon explanation for growth and development, and then you finish the sentence with the roles of ncRNAs in carcinogenesis and cancer development. I recommend you find a more suitable, clearer manner of explaining that ncRNAs are actually involved in both physiological and pathological processes, including cancer-associated hallmarks.

13.  Line 55: “(Human Recombinant Protein (P01).” – there is one “)” missing at the end.

14.  Ling 63: “Studies have shown that two mature types of miR-199a regulate the activities of nor-63 mal cells and participate in corresponding physiological or pathological processes.” – additional references should be provided.

15.  Line 74: “favious tumors” – I think you wanted to write “various tumors”.

16.  Line 75: change “lowly expressed” with “underexpressed”.

17.  Line 76: change “species” with “types”.

18.  Line 77: find a better formulation for “exists as an oncogene.”, especially since you are talking about miRNAs. I recommend “is generally considered an oncomiRNA” and add proper references. 

19.  Line 78: Please rephrase “As the organ with the highest incidence of malignant tumor metastasis, tumors of all tissues and organs in the body can basically metastasize to the lung.” as it is very unclear.

20.  Line 85: change “species” with “types”.

21.  Line 86: It is vert uncommon to start a sentence with “And”, please rephase it and use a suitable connector to the previous sentence but keep it short and clear.

22.  Line 89: the adverb “recently” is incorrectly placed in the sentence.

23.  Line 93: Please provide suitable references for “The expression of miR-199a-5p was significantly decreased in NSCLC tissues and cell 94 lines. Still, the down-regulation mechanism of mir-199a-5p expression and the role of its 95 downstream targeting factors in NSCLC are not fully understood.”.

24.  Line 100: Reference [21] should be placed after mentioning the author “Mudduluru G et al.”.

25.  Line 122: Please provide more than one reference, as you mentioned “Furthermore, recent studies[25]”.

26.  Line 125: Please provide additional references for “Given the available cancer biomarkers, methylation of miRNAs 125 may become an important component of future early cancer detection systems.”

27.  Lines 132-133: Replace “The study” with “One study” or rephrase the whole paragraph so it would make more sense.

28.  Line 150: I recommend using the present continuous tense, so “plays” instead of “played”.

29.  Line 171: Reference [34] should be place at the end of the sentence.

30.  Line 179-180: Please provide references for the first sentence.

31.  Line 190: Please rearrange the Table 1 by miRNA type (miR-199a-5p; miR-199a-3p; miR-199a-5p /3p) and remove the miR-199a example if you can’t exactly find if it is 5p or 3p.

32.  Line 206: Reference 37 should be positioned after the evidence, not after your conclusion based on that evidence.

33.  Line 213: Please provide additional references, other than [18].

34.  Line 219: Please rephrase the first sentence. I recommend removing “discover” and find a more proper expression for “in the face of”. Additional references should be provided as well.

35.  Line 229: Please rearrange the Table 2 by miRNA type (miR-199a-5p; miR-199a-3p; miR-199a-5p /3p) and remove the miR-199a example if you can’t exactly find if it is 5p or 3p. You could consider extending it by adding additional evidence.

36.  Line 313: Since you have already added the abbreviation “EVs” for extracellular vesicles, please stick with it.

37.  Line 327: Please provide additional references, other than [80].

38.  Line 337: Please rearrange the Table 3 by miRNA type (miR-199a-5p; miR-199a-3p; miR-199a-5p /3p) and remove the miR-199a example if you can’t exactly find if it is 5p or 3p.

39.  Line 352: Please provide additional reference for the carcinogenic role of miR-199a-5p.

40.  Line 362: Please revise the sentence so it won’t start with “And” or use a suitable connector to link it with the previous one.

41.  Line 375: Please provide additional reference for the statement “miR-199a-3p was a target of TUG1.”.

42.  Line 377: Please add additional references, other than [90].

43.  Line 390: Please add additional references, other than [93].

44.  Line 399: Please rearrange the Table 4 by miRNA type (miR-199a-5p; miR-199a-3p; miR-199a-5p /3p).

45.  Line 408: Please add additional references, other than [95].

46.  Line 432: Please rearrange the Table 5 by miRNA type (miR-199a-5p; miR-199a-3p; miR-199a-5p /3p) and remove the miR-199a example if you can’t exactly find if it is 5p or 3p.

47.  Lines 435-437: Please provide suitable references for “In addition to the above lncRNAs and circRNAs, there is also the relationship between tRFs and miRNAs. TRFs are not random by-products of tRNA degradation but have precise sequence structures, specific expression patterns, and specific biological effects. MiRNAs sequences are found within tRFs.

48.  Line 455: Please provide additional references, other than [106].

49.  Line 476: Please rephrase “In the diagnosis of lung cancer, the diagnosis is often delayed.” so you won’t need to use diagnosis twice as it is disturbing for the reader.

50.  Line 477: This part is no clear enough “and often the latter.”. Please rephrase.

51.  Line 484: Please provide references.

52.  Line 496: Please provide references.

53.  Line 506: Please do not start the sentence with “And”.

54.  Lines 506-548: This paragraph is way too long. Please divide it in shorter paragraphs so it will be easier to follow.

55.  A dedicated chapter for conclusions could provide added value to this literature review. 

Reviewer 2 Report

Dear authors, 

Thanks for your contribution on this field. This is an interesting state of art on of art on MiR-199a and non-coding RNAs interplay in lung cancer which I already review some time ago.  

You already answered to my raised points about missing arguments and comments in the first submission. 

The manuscript is still well written, organized and suitable for publication. 

Regards                                                                        

Round 2

Reviewer 1 Report

After carefully revising the updated version of the manuscript I agree that significant progresses have been made that increase the clearness of the paper and overall quality. 

Meng et al. have made a great job in describing the role of miR-199a in lung cancer pathogenesis and role in therapy resistance. 

This manuscript is a resubmission of an earlier submission. The following is a list of the peer review reports and author responses from that submission.

Round 1

Reviewer 1 Report

Dear authors,

Thanks for your contribution on this field. This is an interesting state of art on of art on MiR-199a and non-coding RNAs interplay in lung cancer. The manuscript is well written, but I feel a bit disappointed by its content. Indeed, IJMS is a journal with a high IF, around 6.

For this reason, we can expect more than a list of recent finding on the literature. There are no explanations, no links between recent findings or how they can be interpreted regarding lung cancer.

Some cited articles are dedicated to other types of cancer, and it will be interesting if the authors can give their opinion or insight how it can affect lung cancer.

As it is claimed at the end of the abstract: “It aims to provide new insights into the treatment and prevention of lung cancer”, unfortunately I couldn’t read such insights.

The part “3. Biological roles of miR-199a in lung cancer” must be re-written by taking in accounts the previous comment. The first pat, up to “MiR-199a regulates glucose metabolism in lung cancer”, consists only of small paragraphs highlighting new published articles. It looks more like a list of abstract than a review.

The same append with the part “4. Other miRNAs and other ncRNAs interplay and lung cancer”. The interplay is not really emphasized. The Figure 2 maybe move to this part and used to evidence the interplay.

In advance, thank you for considering these comments,

Regards

Reviewer 2 Report

The paper by Meng et al. aims to present an overview of the roles of miR-199a and other ncRNAs in lung cancer. The manuscript starts by explaining the biogenesis and main functions of miR-199a in lung cancer. Still, it soon loses its focus on lung cancer by presenting related data from different cancer and non-malignant diseases. The overall structure and information presented in the manuscript lack novelty and it doesn't succeed in engaging the reader and in convincing over the main roles of miR-199a in lung cancer. 

 What is intriguing is that chapter 4 is chaotically designed as it either selects randomly miRNAs related to lung cancer, in miR-199a and miRNAs, where authors cite themselves, or it loses completely the contact with lung cancer as in the chapters about lncRNA, circRNA and tRNA. 

Also, the conclusion of the manuscript is a 2-page long manuscript that begins with a long paragraph about lung cancer diagnosis and therapy which is out of the scope of this review, and is ended by a long descriptive paragraph that touches slightly on miR-199a and it fails to draw a clear conclusion. 

Round 2

Reviewer 1 Report

Dear authors,

Thanks for providing a very good revised version of your manuscript. Now it meets the golden standard for publication in ijms. 

It's now a pleasure to read it with you comments and arguments on this subject.

Congratulations and all the best